# Efficient transformer with reinforced position embedding for language models

## Abstract

In this paper, we propose an efficient transformer architecture that uses reinforced positional embedding to obtain superior performance with half the number of encoder decoder layers. We demonstrate that concatenating positional encoding with trainable token embeddings, normalizing across tokens in the token embedding matrix, and using the normalized token embedding matrix as the value of the attention layer improve the training and validation loss and the training time in an encoder-decoder Transformer model for a Portuguese-English translation task with 10 epochs or 12 hours of training across 10 trials. Our method, with roughly a threefold parameter reduction compared to the baseline model, yields a mean training loss of 1.21, a mean validation loss of 1.51, and an average training time of 1352.27 seconds per epoch, surpassing the baseline model with the same embedding dimension that employs addition of positional encoding and token embeddings, which achieves a mean training loss of 1.96, a validation loss of 2.18, and an average training time of 4297.79 seconds per epoch. Additionally, we evaluated our proposed architecture and the baseline across 14 diverse translation datasets from TensorFlow. The results indicate that our method consistently achieves lower or comparable training and validation losses, suggesting enhanced learning efficiency.

## 1 Introduction

In this study, we propose modifications to the encoder-decoder Transformer architecture from Vaswani (2017) to enhance training performance on machine translation tasks using the dataset from Ye et al. (2018). Previous work by Ke et al. Ke et al. (2021) demonstrated that employing separate query and key projection matrices for token embeddings and absolute positional embeddings in the self-attention layer accelerates convergence with respect to validation loss in BERT models Lee & Toutanova (2018). They also indicated that the standard attention mechanism in Transformers might be inefficient due to weak correlations between words and their absolute positions.

We present three modifications to enhance the efficiency and performance of the encoder-decoder Transformer architecture. Firstly, we reason that the addition of the token and positional embedding matrix may cause loss of information. To address this, we concatenate the token and positional embedding matrices before the initial encoder and decoder blocks, as shown in Figure 1 (b). Second, we normalize the token embedding matrix across tokens, as shown in Figure 1 (c). Third, we use the normalized token embedding matrix as the value in the attention layer, as shown in Figure 1 (d).

Specifically, let $m$ denote the token embedding dimension and $n$ the sequence length. The input to the first encoder or decoder block consists of the concatenated token and positional embedding matrices, resulting in a matrix of dimension $n \times 2m$. Each column of the token embedding matrix is normalized to have a mean of zero and a standard deviation of one. In the attention layer, the key and query matrices are of dimension $n \times 2m$, while the value matrix is the normalized token embedding matrix, yielding an $n \times m$ dimension. The output of each attention layer is an $n \times 2m$ matrix, which allows for residual connections that maintain positional information across layers. The feedforward layer processes an input matrix of dimension $n \times 2m$ and outputs a matrix of the same dimension. The modified architecture is shown in Figure 2.

We compare our proposed model against a baseline architecture similar to that of Vaswani (2017) but with double the number of layers and the same embedding dimension. Namely, in the baseline,

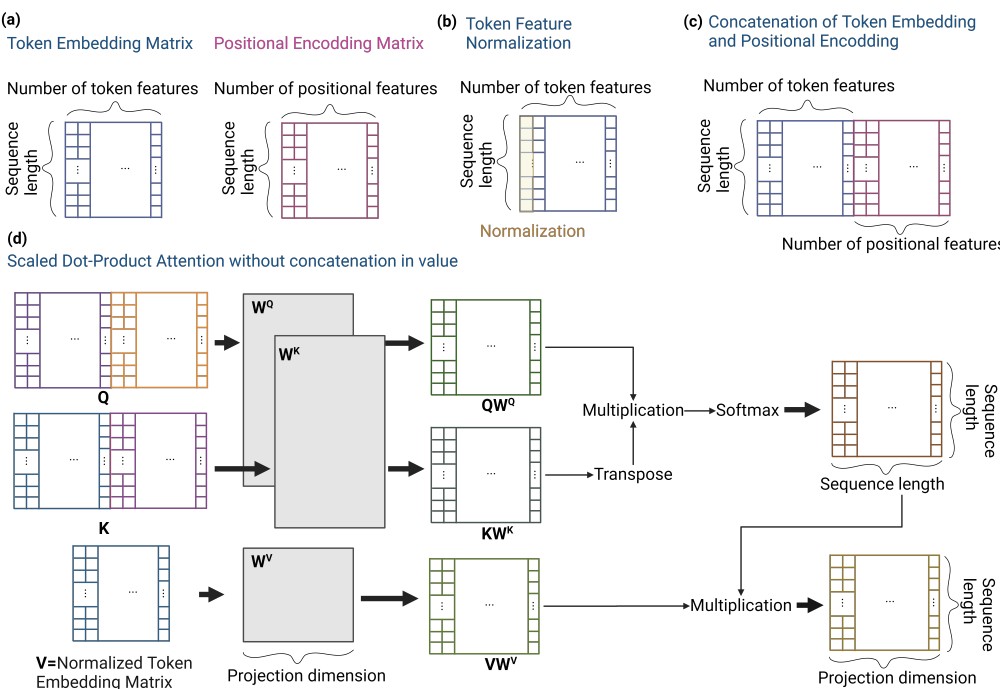

Figure 1: (a) The matrix on the left-hand side is a token embedding matrix with the number of rows the same as the number of tokens and with the number of columns the same as the number of token features for the tokens. Each row in the token embedding matrix corresponds to the token feature vector for the token at that row. The matrix at the right-hand side is a positional embedding matrix with the number of rows the same as the number of tokens and with the number of columns the same as the number of features at that position (row). Each row in the positional embedding matrix is the feature vector correspond to that position (row). (b) We normalize each column of the token embedding matrix to make each column having elements with zero mean and unit variance. (c) After the token feature normalization, we concatenate the positional embedding matrix to the right of the normalized token embedding matrix. (d) For the scaled dot-product attention in each attention layer, the value is the normalized token embedding matrix from the input. Created with BioRender.com.

the token and positional embedding matrices, both of dimension $n \times 2m$, are summed to form the input matrix for the initial encoder or decoder block.

Using the Portuguese-English translation dataset from Ye et al. (2018) and training both models for 10 epochs, our proposed model achieves a training loss of 1.22 and a validation loss of 1.53, outperforming the baseline model, which recorded a training loss of 1.84 and a validation loss of 2.13. Additionally, our proposed model demonstrates an average training time of 1352 seconds per epoch with 2,809,634 parameters, both of which are roughly three times lower than the baseline model's averages of 4298 seconds per epoch and 10,184,162 parameters.

These results indicate that our model not only achieves improved training and validation loss but also demonstrates greater training efficiency compared to the baseline Transformer on the Portuguese-English dataset Ye et al. (2018). To further validate the robustness of our findings across different translation datasets, we applied both models to thirteen additional translation datasets from Ye et al. (2018) training for 10 epochs or 12 hours. Both models successfully completed 10 epochs of training on seven datasets, where our proposed method showed lower training and validation loss in six datasets. However, our method did not yield improved results for the Belarusian-to-English translation dataset, with our model showing a training loss of 4.60 and a validation loss of 4.35, compared to the baseline's training loss of 4.56 and validation loss of 4.30. Notably, our proposed method achieved nearly a threefold reduction in training time compared to the baseline model.

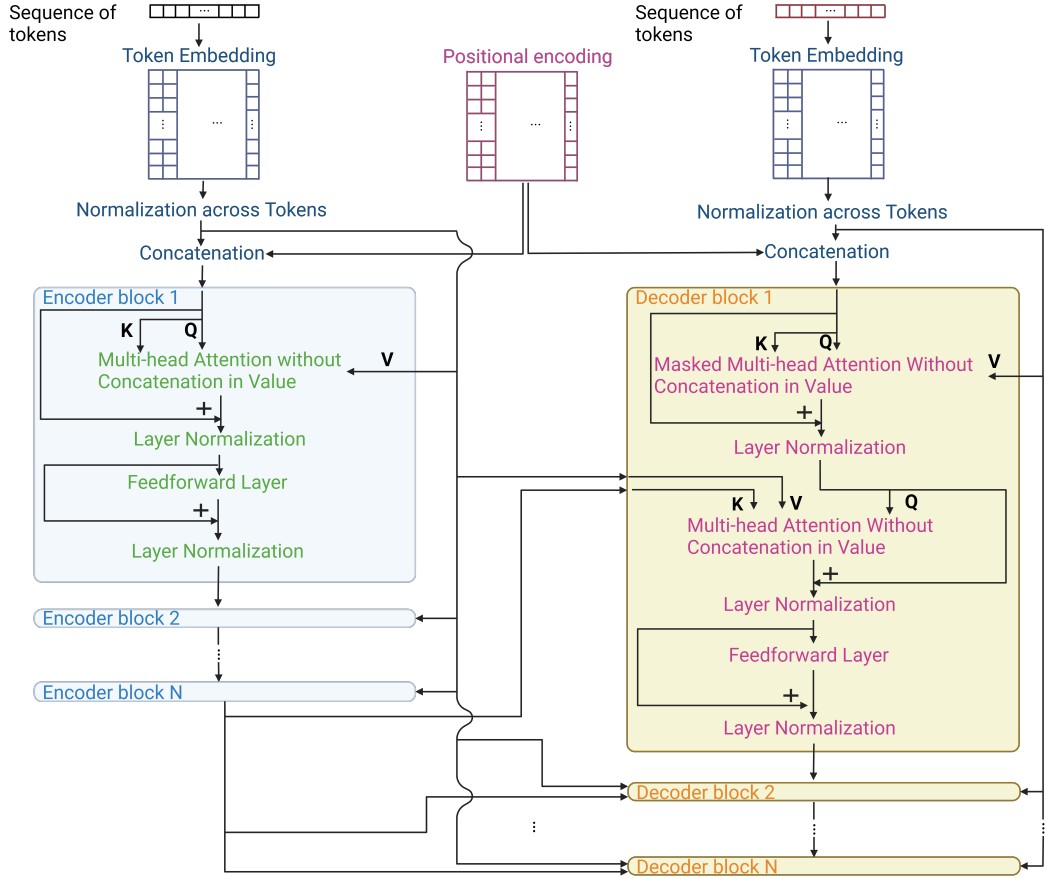

Figure 2: The proposed modified transformer architecture. We made three modification from the transformer architecture in Vaswani (2017). Firstly, each column in the token embedding matrix is normalized to have zero mean and unit variance for both the encoder and decoder. Secondly, The token embedding matrix and the positional embedding matrix is concatenated before the first encoder block and the first decoder block. Lastly, each attention layer has the value without concatenation. Created with BioRender.com.

## 2 MODIFIED TRANSFORMER ARCHITECTURE

Given a sequence $\boldsymbol{x} = \begin{bmatrix} x_1 & x_2 & \cdots & x_n \end{bmatrix} \in \mathbb{R}$ with $n$ elements, the output of a token embedding is a $n \times m$ matrix and it can be written as

$$\boldsymbol{X}_e = \begin{bmatrix} \boldsymbol{e}^{(1)} & \boldsymbol{e}^{(2)} & \cdots & \boldsymbol{e}^{(n)} \end{bmatrix}^T \boldsymbol{W}_1, \tag{1}$$

where $\boldsymbol{W}_1 \in \mathbb{R}^{|Vocab| \times m}$ is an embedding matrix, $|Vocab|$ is the number of tokens in the vocabulary set, $m$ is the dimension of an embedding vector, and $\boldsymbol{e}^{(i)} \in \mathbb{R}^{m \times 1}$ is a unit column vector with a 1 at position $i$ and 0 elsewhere.

Given a sequence $\boldsymbol{y} = \begin{bmatrix} y_1 & y_2 & \cdots & y_n \end{bmatrix} \in \mathbb{R}^n$ with $n$ elements, the output of a token embedding is a $n \times m$ matrix and it can be written as

$$\boldsymbol{Y}_e = \begin{bmatrix} \boldsymbol{e}^{(1)} & \boldsymbol{e}^{(2)} & \cdots & \boldsymbol{e}^{(n)} \end{bmatrix}^T \boldsymbol{W}_2, \tag{2}$$

where $\boldsymbol{W}_2 \in \mathbb{R}^{|Vocab| \times m}$ is an embedding matrix, $|Vocab|$ is the number of tokens in the vocabulary set, $m$ is the dimension of an embedding vector, and $\boldsymbol{e}^{(i)} \in \mathbb{R}^{m \times 1}$ is a unit column vector with a 1 at position $i$ and 0 elsewhere.

For the positional encoding, given the sequence length $n$ and the embedding dimension $m$, the output is $\boldsymbol{P} \in \mathbb{R}^{n \times m}$, where

$$\boldsymbol{P}_{i,j} = \begin{cases} sin(i/(10000^{2j/m})) & \text{if } 1 \leq j \leq m/2 \\ cos(i/(10000^{2j/m})) & \text{if } m/2 + 1 \leq j \leq m \end{cases} \tag{3}$$

for $1 \leq i \leq n$ and $1 \leq j \leq m$.

## 2.1 Normalization across tokens in token embedding

Given a matrix $\boldsymbol{X} \in \mathbb{R}^{n \times m}$, the output of a function for normalization across tokens can be written as

$$TN(\boldsymbol{X}) = \begin{bmatrix} \boldsymbol{x}_1 & \boldsymbol{x}_2 & \cdots & \boldsymbol{x}_m \end{bmatrix}, \tag{4}$$

where

$$\boldsymbol{x}_j = \frac{\boldsymbol{X}_{:,j} - E[\boldsymbol{X}_{:,j}]}{\sqrt{Var[\boldsymbol{X}_{:,j}]}}, \tag{5}$$

$$E[\boldsymbol{X}_{:,j}] = \frac{1}{n} \sum_{i=1}^{n} \boldsymbol{X}_{i,j}, \tag{6}$$

$$Var[\boldsymbol{X}_{:,j}] = E[(\boldsymbol{X}_{:,j} - E[\boldsymbol{X}_{:,j}])^2], \tag{7}$$

and $\boldsymbol{X}_{:,j}$ is the $j$-th column of matrix $\boldsymbol{X}$ for $1 \leq i \leq n$ and $1 \leq j \leq m$.

First, we propose to normalize across tokens in the token embedding matrix $\boldsymbol{X}_e$ and $\boldsymbol{Y}_e$ such that

$$\bar{\boldsymbol{X}}_e = TN(\boldsymbol{X}_e) \tag{8}$$

and

$$\bar{\boldsymbol{Y}}_e = TN(\boldsymbol{Y}_e). \tag{9}$$

## 2.2 Concatenation between token embedding and positional embedding

Second, we propose to concatenate the output of the token embedding and the output of the positional encoding such that

$$\boldsymbol{X}_I^{concat} = \begin{bmatrix} \bar{\boldsymbol{X}}_e & \boldsymbol{P} \end{bmatrix} \tag{10}$$

and

$$\boldsymbol{Y}_I^{concat} = \begin{bmatrix} \bar{\boldsymbol{Y}}_e & \boldsymbol{P} \end{bmatrix} \tag{11}$$

replace the $\boldsymbol{X}_I$ and $\boldsymbol{Y}_I$ in (28) and (38) in the baseline model in Appendix A, respectively.

## 2.3 Avoid concatenation in value in all the attention layers

Third, we propose to use the normalized token embedding matrix in value in the multi-head attention, multi-head cross attention, and masked multi-head attention defined in (29), (41), and (39) in the Appendix A, respectively.

Given three matrices $\boldsymbol{Q}, \boldsymbol{K} \in \mathbb{R}^{n \times 2m}$ and $\boldsymbol{V} \in \mathbb{R}^{n \times m}$, the scaled dot-product attention is defined as

$$Attention(\boldsymbol{Q}, \boldsymbol{K}, \boldsymbol{V}) = softmax(\frac{\boldsymbol{Q}\boldsymbol{K}^T}{\sqrt{m}})\boldsymbol{V}, \tag{12}$$

where $softmax : \mathbb{R}^{n \times s} \to [0,1]^{n \times s}$ for $s \in \mathbb{N}^+$ is a Softmax function with $softmax(\boldsymbol{A})_{i,j} = \frac{\boldsymbol{A}_{i,j}}{\sum_{j=1}^{s} e^{\boldsymbol{A}_{i,j}}}$ for $\boldsymbol{A} \in \mathbb{R}^{n \times s}$. The masked scaled dot-product attention is defined as

$$MaskedAttention(\boldsymbol{Q}, \boldsymbol{K}, \boldsymbol{V}) = softmax(\boldsymbol{M} + \frac{\boldsymbol{Q}\boldsymbol{K}^T}{\sqrt{m}})\boldsymbol{V}. \tag{13}$$

where $\boldsymbol{M} \in \mathbb{R}^{n \times n}$ is a matrix with $\boldsymbol{M}_{i,j} = \begin{cases} \infty & \text{if } j > i \\ 0 & \text{otherwise} \end{cases}$ for $1 \leq i \leq n$ and $1 \leq j \leq n$.

For the proposed multi-head attention layer without concatenation in value in the $u$-th encoder block, given three matrices $\boldsymbol{Q}_1, \boldsymbol{K}_1 \in \mathbb{R}^{n \times 2m}$ and $\boldsymbol{V}_1 \in \mathbb{R}^{n \times m}$, the output is computed by

$$PMHA(\boldsymbol{Q}_1, \boldsymbol{K}_1, \boldsymbol{V}_1, u) = [\boldsymbol{H}_1 \quad \boldsymbol{H}_2 \quad \cdots \quad \boldsymbol{H}_p] \boldsymbol{W}^{O_u}, \tag{14}$$

where

$$\boldsymbol{H}_k = Attention(\boldsymbol{Q}_1 \boldsymbol{W}_k^{Q_1}, \boldsymbol{K}_1 \boldsymbol{W}_k^{K_1}, \boldsymbol{V}_1 \boldsymbol{W}_k^{V_1}), \tag{15}$$

$u = 1, 2, \cdots, N$, $k = 1, 2, \cdots, p$, $N$ is the total number of encoders, $p$ is the number of heads, $\boldsymbol{W}^{O_u} \in \mathbb{R}^{qp \times 2m}$, $\boldsymbol{W}_k^{Q_1} \in \mathbb{R}^{2m \times r}$, $\boldsymbol{W}_k^{K_1} \in \mathbb{R}^{2m \times r}$, $\boldsymbol{W}_k^{V_1} \in \mathbb{R}^{m \times q}$, $r$ is the dimension of the query and key projection, $q$ is the dimension of the value projection.

For the proposed masked multi-head attention layer without concatenation in value in the $u$-th decoder block, given three matrices $\boldsymbol{Q}_2, \boldsymbol{K}_2 \in \mathbb{R}^{n \times 2m}$ and $\boldsymbol{V}_2 \in \mathbb{R}^{n \times m}$, the output is computed by

$$PMMHA(\boldsymbol{Q}_2, \boldsymbol{K}_2, \boldsymbol{V}_2, u) = \begin{bmatrix} \boldsymbol{H}_{masked_1} & \boldsymbol{H}_{masked_2} & \cdots & \boldsymbol{H}_{masked_p} \end{bmatrix} \boldsymbol{W}_{masked}^{O_u}, \tag{16}$$

where

$$\boldsymbol{H}_{masked_k} = MaskedAttention(\boldsymbol{Q}_2 \boldsymbol{W}_k^{Q_2}, \boldsymbol{K}_2 \boldsymbol{W}_k^{K_2}, \boldsymbol{V}_2 \boldsymbol{W}_k^{V_2}), \tag{17}$$

$u = 1, 2, \cdots, N$, $k = 1, 2, \cdots, p$, $N$ is the total number of decoders, $p$ is the number of heads, $\boldsymbol{W}_{masked}^{O_u} \in \mathbb{R}^{qp \times 2m}$, $\boldsymbol{W}_k^{Q_2} \in \mathbb{R}^{2m \times r}$, $\boldsymbol{W}_k^{K_2} \in \mathbb{R}^{2m \times r}$, $\boldsymbol{W}_k^{V_2} \in \mathbb{R}^{m \times q}$, $r$ is the dimension of the query and key projection, and $q$ is the dimension of the value projection.

For the proposed multi-head cross attention layer without concatenation in value in the $u$-th decoder block, given three matrices $\boldsymbol{Q}_3, \boldsymbol{K}_3 \in \mathbb{R}^{n \times 2m}$ and $\boldsymbol{V}_3 \in \mathbb{R}^{n \times m}$, the output is computed as

$$PMHCA(\boldsymbol{Q}_3, \boldsymbol{K}_3, \boldsymbol{V}_3, u) = \begin{bmatrix} \boldsymbol{H}_{cross_1} & \boldsymbol{H}_{cross_2} & \cdots & \boldsymbol{H}_{cross_p} \end{bmatrix} \boldsymbol{W}_{causal}^{O_u}, \tag{18}$$

where

$$\boldsymbol{H}_{cross_k} = Attention(\boldsymbol{Q}_3 \boldsymbol{W}_k^{Q_3}, \boldsymbol{K}_3 \boldsymbol{W}_k^{K_3}, \boldsymbol{V}_3 \boldsymbol{W}_k^{V_3}), \tag{19}$$

$u = 1, 2, \cdots, N$, $k = 1, 2, \cdots, p$, $N$ is the total number of decoders, $p$ is the number of heads, $\boldsymbol{W}^{O_u} \in \mathbb{R}^{qp \times 2m}$, $\boldsymbol{W}_k^{Q_3} \in \mathbb{R}^{2m \times r}$, $\boldsymbol{W}_k^{K_3} \in \mathbb{R}^{2m \times r}$, $\boldsymbol{W}_k^{V_3} \in \mathbb{R}^{m \times q}$, $r$ is the dimension of the query and key projection, $q$ is the dimension of the value projection.

## 2.4 THE OUTPUT OF EACH ENCODER AND DECODER BLOCK IN THE MODIFIED TRANSFORMER MODEL

The resulting output for the first encoder block and the $N$ stacks of encoder blocks are

$$\boldsymbol{X}_{E_1}^{concat} = LN(FFN(\boldsymbol{X}_{M_1}^{concat}) + \boldsymbol{X}_{M_1}^{concat}) \tag{20}$$

and

$$\boldsymbol{X}_{E_N}^{concat} = LN(FFN(\boldsymbol{X}_{M_N}^{concat}) + \boldsymbol{X}_{M_N}^{concat}), \tag{21}$$

respectively, where $N > 1$, $\boldsymbol{X}_{M_1}^{concat} = LN(PMHA(\boldsymbol{X}_I^{concat}, \boldsymbol{X}_I^{concat}, \bar{\boldsymbol{X}}_e, 1) + \boldsymbol{X}_I^{concat})$, $\boldsymbol{X}_{M_N}^{concat} = LN(PMHA(\boldsymbol{Q}, \boldsymbol{K}, \boldsymbol{V}, N) + \boldsymbol{X}_{E_{N-1}^{concat}})$, $\boldsymbol{Q} = \boldsymbol{K} = \boldsymbol{X}_{E_{N-1}^{concat}}$, $\boldsymbol{V} = \bar{\boldsymbol{X}}_e$, $LN$ and $FFN$ are the layer normalization and the feed forward network defined in (32), (31) in Appendix A, respectively, and $PMHA$ is the proposed multi-head attention defined in (14).

The resulting output for the first decoder block and the $N$ stacks of decoder blocks becomes

$$\boldsymbol{Y}_{D_1}^{concat} = LN(FFN(\boldsymbol{Y}_{C_1}^{concat}) + \boldsymbol{Y}_{C_1}^{concat}) \tag{22}$$

and

$$\boldsymbol{Y}_{D_N}^{concat} = LN(FFN(\boldsymbol{Y}_{C_N}^{concat}) + \boldsymbol{Y}_{C_N}^{concat}), \tag{23}$$

respectively, where $\boldsymbol{Y}_{C_1}^{concat} = LN(PMHCA(\boldsymbol{Y}_{M_1}^{concat}, \boldsymbol{X}_{E_N}^{concat}, \bar{\boldsymbol{X}}_e, 1) + \boldsymbol{Y}_{M_1}^{concat})$, $\boldsymbol{Y}_{M_1}^{concat} = LN(PMMHA(\boldsymbol{Y}_I^{concat}, \boldsymbol{Y}_I^{concat}, \bar{\boldsymbol{Y}}_e, 1) + \boldsymbol{Y}_I^{concat})$, $\boldsymbol{Y}_{C_N}^{concat} = LN(PMHCA(\boldsymbol{Y}_{M_N}^{concat}, \boldsymbol{X}_{E_N}^{concat}, \bar{\boldsymbol{X}}_e, N) + \boldsymbol{Y}_{M_N}^{concat})$, $\boldsymbol{Y}_{M_N}^{concat} = LN(PMMHA(\boldsymbol{Y}_{D_{N-1}}^{concat}, \boldsymbol{Y}_{D_{N-1}}^{concat}, \bar{\boldsymbol{Y}}_e, N) + \boldsymbol{Y}_{D_{N-1}}^{concat})$, and $PMHCA$ and $PMMHA$ are the proposed multi-head cross attention and the proposed masked multi-head attention defined in (18) and (16), respectively.

## 2.5 The output layer of the modified transformer model

The output layer of the proposed transformer model composed of a feed forward network and a softmax layer, which can be written as

$$\boldsymbol{T}^{concat} = softmax(\boldsymbol{Y}_{D_N}^{concat}\boldsymbol{W}^{final}), \qquad (24)$$

where $\boldsymbol{W}^{final} \in \mathbb{R}^{2m \times |Vocab|}$. The $i$-th predicted token is the token corresponds to the $j$-th column that has the maximum value among all the elements in the $i$-th row of $\boldsymbol{T}^{concat}$.

## 3 Experiments and results

### 3.1 Experimental setup

We first evaluated our proposed model and the baseline model using the Portuguese-English translation dataset from TensorFlow Abadi et al. (2015) for 10 epochs or 12 hours of training across 10 trials. The dataset contains 51,785 training, 1,193 validation, and 1,803 test pairs of Portuguese and English text. The encoder's input is the Portuguese text, and the decoder's input is the English text, with both sequences capped at 128 tokens. The target labels are the decoder inputs shifted right by one token. We applied wordpiece tokenization for the Portuguese-English translation dataset from TensorFlow Abadi et al. (2015). For the second experiment, we evaluated our proposed model and the baseline model using the Portuguese-English translation dataset from TensorFlow Abadi et al. (2015) for 10 epochs of training or 12 hours in one trial. We applied wordpiece tokenization for each dataset from TensorFlow Abadi et al. (2015).

The model was trained using the Adam optimizer from Vaswani (2017):

$$lr(N_s) = m^{-0.5} min(N_s^{-0.5}, N_s N_w^{-1.5}), \qquad (25)$$

where $m$ is the embedding dimension, $N_s \in \mathbb{Z}$ is the current training step, and $N_w = 4000$ is called the warm-up step. The loss function is a masked cross-entropy loss. The training epoch is 10. The drop-out rate is 0.1. The batch size is 64.

For our proposed model (Section 2.2), we used 2 encoder and decoder blocks, the token embedding dimension of $m = 64$, 256 hidden neurons in the feed-forward network, 4 attention heads, and the projection dimension of 64 for query, key, and value. The proposed model contains 2,809,634 parameters.

For the baseline model, the number of encoder and decoder blocks is 4; the token embedding dimension is $m = 128$; the number of hidden neurons in the feed-forward network is 512; the number of attention heads is $p = 8$; the projection dimension is 128 for query, key, and value. The baseline model contains 10,184,162 parameters, which is roughly three times more than the proposed model.

### 3.2 Experimental results and discussion

We trained both the baseline and proposed models for 10 epochs or 12 hours across 10 different trials on two A100 40GB GPUs on the Portuguese to English translation dataset from Ye et al. (2018). For the baseline model, the 10 epochs of training were completed for 4 trails; 9 epochs of training is completed for 4 trails; and 8 epochs of training were completed for 2 trails. For the proposed model, 10 epochs of training were completed for all the trials. The comparison of the learning curves between the baseline and the proposed method are shown in Figure 3 (a). The mean training losses of the baseline are roughly 6.60, 4.55, 3.82, 3.29, 2.89, 2.57, 2.30, 2.11, 1.96, and 1.84. The mean training losses of the model from the proposed method for each epoch are roughly 6.68, 4.55, 3.62, 2.89, 2.36, 1.97, 1.67, 1.46, 1.32, and 1.21, which are less than the mean training losses of the baseline after 3 epochs. The mean validation losses of the baseline are roughly 5.04, 4.06, 3.46, 3.01, 2.77, 2.49, 2.35, 2.25, 2.18, and 2.14. The mean validation losses of the model from the proposed method for each epoch are roughly 5.08, 3.93, 3.10, 2.51, 2.17, 1.88, 1.72, 1.63, 1.56, and 1.51, which are less than the mean validation losses of the baseline after 2 epochs. The comparison from the learning curves shows that the improvement of the training loss and validation loss of our model compared to the baseline is consistent for 10 different trials.

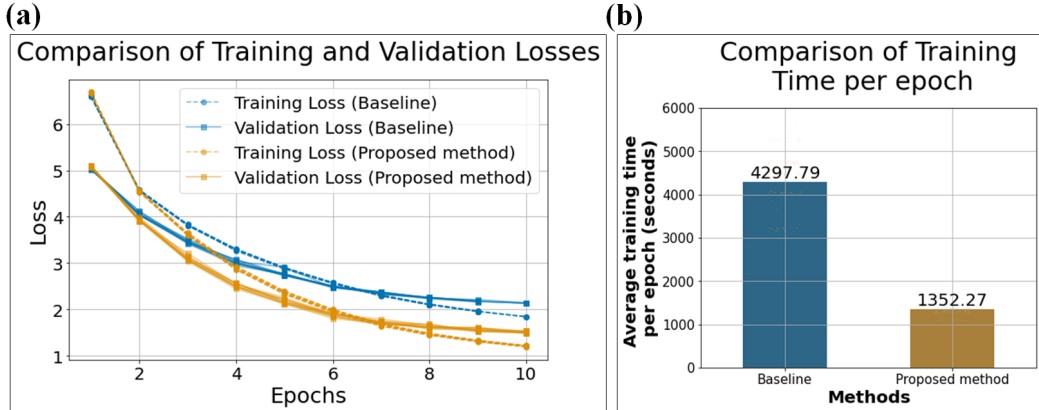

Figure 3: (a) The transparent blue dashed lines and the transparent blue solid lines are the training loss and the validation loss of the baseline transformer model on the Portuguese to English translation dataset from Ye et al. (2018), respectively. The transparent brown dashed lines and the transparent brown solid lines are the training loss and the validation loss of the proposed transformer model on the Portuguese to English translation dataset from Ye et al. (2018), respectively. Each line represent the training or validation loss for one trial. Each model are trained for 10 epochs or 12 hours. The baseline model has 4 trials that finished 10 epochs of training; 4 trails finished 9 epochs of training; 2 trials finished 8 epochs of training. The proposed model finished 10 epochs of training for all the trials. The training losses of the baseline have the mean of roughly 6.60, 4.55, 3.82, 3.29, 2.89, 2.57, 2.30, 2.11, 1.96, and 1.84 and the mean of the validation losses of the baseline are roughly 5.04, 4.06, 3.46, 3.01, 2.77, 2.49, 2.35, 2.25, and 2.18 for 10 different trials. The training losses of the proposed model have the mean of roughly 6.68, 4.55, 3.62, 2.89, 2.36, 1.97, 1.67, 1.46, 1.32, and 1.21 and the mean of the validation losses of the proposed model are roughly 5.08, 3.93, 3.10, 2.51, 2.17, 1.88, 1.72, 1.63, 1.56, and 1.51 for 10 different trials. The proposed model shows a lower mean of training losses after 3 epochs and a lower mean of validation losses after 2 epochs. (b) The bar plot shows the average training time per epoch for both the baseline and the proposed model on the Portuguese to English translation dataset from Ye et al. (2018). The average training time per epoch for the baseline is roughly 4297.79 seconds, which are higher than roughly 1352.27 seconds for the proposed model. In addition, The variance of the training time for the baseline is roughly 675.79 seconds, which are higher than roughly 144.50 seconds for the proposed model.

The average training time for the baseline model and the proposed model for 10 epochs or 12 hours and across 10 different trials is shown in Figure 3 (b). The average training time and the standard deviation of the training time for the proposed model are roughly 1352.27 seconds and 144.50 seconds, respectively, which are less than those in the baseline model with the average training time of 4297.79 seconds and the standard deviation of the training time of 675.79 seconds. The proposed model shows roughly a threefold reduction in training time compared to the baseline. This may be due to the approximately threefold reduction in the number of parameters in the proposed model, which has 2,809,634 parameters, compared to the baseline with 10,184,162 parameters. In summary, the results from Figure 3 show that the proposed model improves the training and validation loss while achieving a threefold reduction in training time.

To verify that the improvement of the training and validation loss and the improvement for the training time are consistent for different translation task, we compared the baseline the the proposed model after training of 10 epochs or 12 hours for one trial on fourteen different translation datasets from Ye et al. (2018). The number of training data and validation data for each dataset are show in the second and third column in Table 1, respectively. The results from the baseline are shown from the fourth to the seventh column and the results from the proposed model are shown from the eighth to eleventh column in Table 1. The fourth and the eighth columns show the number of epoch the corresponding model has completed in 12 hours. The training and validation loss of the baseline at the epoch indicated in the fourth column are shown in the fifth and the sixth column in Table 1. The training and validation loss of the proposed model at the epoch indicated in the fourth column are shown in the ninth and the tenth column in Table 1. The seventh and the the

Table 1: Comparison of baseline and proposed transformer architectures

| Dataset | Data (No.) | | Baseline | | | | Proposed method | | | |
|---------|------------|-----|----------|----------|----------|-----------|-----|----------|----------|-----------|
| | Train. | Val. | Ep. | Train. loss | Val. loss | Comp. Time (s) | Ep. | Train. loss | Val. loss | Comp. Time (s) |
| az_to_en | 5946 | 671 | 10 | 4.34 | 4.23 | 702 | 10 | **4.32** | **4.22** | **164** |
| aztr_to_en | 188396 | 671 | 2 | 3.59 | 3.47 | 15629 | 6 | **1.69** | **1.79** | **5628** |
| be_to_en | 4509 | 248 | 10 | 4.56 | 4.30 | 573 | 10 | 4.60 | 4.35 | **114** |
| beru_to_en | 212614 | 248 | 1 | 4.89 | 3.88 | 22045 | 6 | **1.64** | **1.68** | **6856** |
| es_to_pt | 44938 | 1016 | 10 | 2.15 | 2.41 | 3341 | 10 | **1.46** | **1.75** | **1157** |
| fr_to_pt | 43873 | 1131 | 10 | 2.33 | 2.80 | 3360 | 10 | **1.62** | **2.05** | **1139** |
| gl_to_en | 10017 | 682 | 10 | 3.77 | 3.75 | 1089 | 10 | **3.69** | **3.67** | **256** |
| glpt_to_en | 61802 | 682 | 9 | 1.92 | 2.48 | 4502 | 10 | **1.17** | **1.80** | **1644** |
| he_to_pt | 48511 | 1145 | 10 | 2.57 | 3.00 | 2970 | 10 | **1.68** | **2.07** | **1263** |
| it_to_pt | 46259 | 1162 | 9 | 2.45 | 2.66 | 4335 | 10 | **1.54** | **1.80** | **1258** |
| pt_to_en | 51785 | 1193 | 10 | 1.84 | 2.13 | 4049 | 10 | **1.22** | **1.53** | **1345** |
| ru_to_en | 208106 | 4805 | 3 | 2.94 | 2.70 | 11055 | 7 | **1.55** | **1.51** | **6122** |
| ru_to_pt | 47278 | 1184 | 8 | 3.22 | 3.43 | 5004 | 10 | **1.87** | **2.23** | **845** |
| tr_to_en | 182450 | 4045 | 2 | 3.60 | 3.03 | 19445 | 6 | **1.66** | **1.57** | **4971** |

Abbreviations: No.: Number; Train.: Training; Val.: Validation; Ep.: Epochs; Comp. Time (s): Computational Time (seconds); az: Azerbaijani; en: English; tr: Turkish; be: Belarusian; ru: Russian; es: Spanish; pt: Portuguese; fr: French; gl: Galician; he: Hebrew; it: Italian; tr: Turkish.

eleventh columns show the average computational time per epoch in seconds for the baseline and the proposed model in Table 1. The result in Table 1 shows that the proposed model achieved an improved training and validation loss compared to the baseline for 13 different translation datasets except for the Belarusian to English translation dataset (be_to_en). For the Belarusian to English translation dataset, the baseline shows a training loss of 4.56 and a validation loss of 4.30, which are lower than those in the proposed model with a training loss of 4.60 and 4.35. However, it is important to note that the proposed model outperformed the baseline in the majority of other thirteen datasets, achieving a reduction in training and validation loss. In addition, the training time for the proposed method is roughly a two to four times reduction from the training time of the baseline model for all the fourteen datasets.

# 4 CONCLUSION

The proposed Transformer architecture, which normalizes the token embedding matrix to have zero mean and unit variance, concatenates token embeddings with positional embeddings before the first encoder and decoder blocks, and uses the normalized token embedding matrix as the value in the attention layer, demonstrates improvements in training loss, validation loss, and computational time across thirteen translation datasets from Ye et al. (2018) compared to the baseline model with the same embedding dimension but with double the number of encoder and decoder blocks. The proposed model achieves a training loss with a mean of roughly 1.21, a validation loss with a mean of roughly 1.51, an average training time of 1352.27 seconds per epoch for 10 epochs of training across 10 trials, which are all lower than those from the baseline which have a training loss with a mean of roughly 1.96, a validation loss with a mean of roughly 2.18, an average training time of 4297.79 seconds per epoch for 10 epochs or 12 hours of training across 10 trials on the Portuguese to English translation dataset from Ye et al. (2018). The proposed model shows a lower training and validation loss compared to the baseline on thirteen out of fourteen different translation datasets from Ye et al. (2018) and shows around two to four times reduction on the training time on all the fourteen datasets from Ye et al. (2018). The proposed model shows promising improvements on thirteen out of fourteen datasets, suggesting that its improvements over the baseline may be robust across a range of translation tasks.

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

## A BASELINE TRANSFORMER ARCHITECTURE

We introduce the architecture of the baseline Transformer we used for showing the improvement of our proposed model. The baseline model architecture is similar to the Transformer architecture in the "Neural machine translation with a Transformer and Keras" tutorial from the TensorFlow Abadi et al. (2015) website.

### A.1 ENCODER ARCHITECTURE IN THE BASELINE TRANSFORMER

Given a sequence $\boldsymbol{x} = [x_1 \quad x_2 \quad \cdots \quad x_n] \in \mathbb{R}$ with $n$ elements, the output of a token embedding is a $n \times m$ matrix and it can be written as

$$\boldsymbol{X}_e = \begin{bmatrix} \boldsymbol{e}^{(1)} & \boldsymbol{e}^{(2)} & \cdots & \boldsymbol{e}^{(n)} \end{bmatrix}^T \boldsymbol{W}_1, \tag{26}$$

where $\boldsymbol{W}_1 \in \mathbb{R}^{|Vocab| \times m}$ is an embedding matrix, $|Vocab|$ is the number of tokens in the vocabulary set, $m$ is the dimension of an embedding vector, and $\boldsymbol{e}^{(i)} \in \mathbb{R}^{m \times 1}$ is a unit column vector with a 1 at position $i$ and 0 elsewhere.

For the positional encoding, given the sequence length $n$ and the embedding dimension $m$, the output is $\boldsymbol{P} \in \mathbb{R}^{n \times m}$, where

$$\boldsymbol{P}_{i,j} = \begin{cases} sin(i/(10000^{2j/m})) & \text{if } 1 \leq j \leq m/2 \\ cos(i/(10000^{2j/m})) & \text{if } m/2 + 1 \leq j \leq m \end{cases} \tag{27}$$

for $1 \leq i \leq n$ and $1 \leq j \leq m$.

The input for the first encoder block is

$$\boldsymbol{X}_I = \frac{1}{\sqrt{m}} \boldsymbol{X}_e + \boldsymbol{P}. \tag{28}$$

For each encoder block, it contains a multi-head attention layer and a feed forward network layer. For the multi-head attention layer in the $u$-th encoder block, given three matrices $\boldsymbol{Q}_1, \boldsymbol{K}_1, \boldsymbol{V}_1 \in \mathbb{R}^{n \times m}$, the output is computed by

$$MHA(\boldsymbol{Q}_1, \boldsymbol{K}_1, \boldsymbol{V}_1, u) = \begin{bmatrix} \boldsymbol{H}_1 & \boldsymbol{H}_2 & \cdots & \boldsymbol{H}_p \end{bmatrix} \boldsymbol{W}^{O_u}, \tag{29}$$

where

$$\boldsymbol{H}_k = softmax(\frac{\boldsymbol{Q}_1 \boldsymbol{W}_k^{Q_1}(\boldsymbol{K}_1 \boldsymbol{W}_k^{K_1})^T}{\sqrt{m}})\boldsymbol{V}_1 \boldsymbol{W}_k^{V_1}, \tag{30}$$

$u = 1, 2, \cdots, N$, $k = 1, 2, \cdots, p$, $N$ is the total number of encoders, $p$ is the number of heads, $\boldsymbol{W}^{O_u} \in \mathbb{R}^{qp \times m}$, $\boldsymbol{W}_k^{Q_1} \in \mathbb{R}^{m \times r}$, $\boldsymbol{W}_k^{K_1} \in \mathbb{R}^{m \times r}$, $\boldsymbol{W}_k^{V_1} \in \mathbb{R}^{m \times q}$, $r$ is the dimension of the query and key projection, $q$ is the dimension of the value projection, and $softmax : \mathbb{R}^{n \times s} \to [0, 1]^{n \times s}$ for $s \in \mathbb{N}^+$ is a Softmax function with $softmax(\boldsymbol{A})_{i,j} = \frac{\boldsymbol{A}_{i,j}}{\sum_{j=1}^s e^{\boldsymbol{A}_{i,j}}}$ for $\boldsymbol{A} \in \mathbb{R}^{n \times s}$.

For the feed forward layer in the $u$-th encoder block, given a matrix $\boldsymbol{X}_f \in \mathbb{R}^{n \times m}$, the output of the feed forward layer is

$$FFN(\boldsymbol{X}_f, u) = ReLU(\boldsymbol{X}_f \boldsymbol{W}_{2_u} + \boldsymbol{B}_{1_u})\boldsymbol{W}_{3_u} + \boldsymbol{B}_{2_u}, \tag{31}$$

where $ReLU$ is an element-wise rectified linear unit (ReLU) activation function $ReLU(\boldsymbol{A}_{i,j}) = max(0, \boldsymbol{A}_{i,j})$ for $\boldsymbol{A} \in \mathbb{R}^{n \times m}$, $\boldsymbol{W}_{2_u} \in \mathbb{R}^{m \times s}$, $\boldsymbol{W}_{3_u} \in \mathbb{R}^{s \times m}$, $\boldsymbol{B}_{1_u} \in \mathbb{R}^{n \times s}$, and $\boldsymbol{B}_{2_u} \in \mathbb{R}^{n \times m}$.

Given a matrix $\boldsymbol{X} \in \mathbb{R}^{n \times m}$, the output of a function for layer normalization can be written as

$$LN(\boldsymbol{X}) = \begin{bmatrix} \boldsymbol{x}_1 & \boldsymbol{x}_2 & \cdots & \boldsymbol{x}_n \end{bmatrix}^T, \tag{32}$$

where

$$\boldsymbol{x}_i = (\frac{\boldsymbol{X}_{i,:} - E[\boldsymbol{X}_{i,:}]}{\sqrt{Var[\boldsymbol{X}_{i,:}]}})^T, \tag{33}$$

$$E[\boldsymbol{X}_{i,:}] = \frac{1}{m}\sum_{j=1}^m \boldsymbol{X}_{i,j}, \tag{34}$$

$$Var[\boldsymbol{X}_{i,:}] = E[(\boldsymbol{X}_{i,:} - E[\boldsymbol{X}_{i,:}])^2], \tag{35}$$

and $\boldsymbol{X}_{i,:}$ is the $i$-th row of matrix $\boldsymbol{X}$ for $1 \leq i \leq n$ and $1 \leq j \leq m$.

The output of the first encoder block can be written as

$$\boldsymbol{X}_{E_1} = LN(FFN(\boldsymbol{X}_{M_1}) + \boldsymbol{X}_{M_1}), \tag{36}$$

where $\boldsymbol{X}_{M_1} = LN(MHA(\boldsymbol{X}_I, \boldsymbol{X}_I, \boldsymbol{X}_I, 1) + \boldsymbol{X}_I)$.

The output of the $N$ stacks of encoder blocks can be written as

$$\boldsymbol{X}_{E_N} = LN(FFN(\boldsymbol{X}_{M_N}) + \boldsymbol{X}_{M_N}), \tag{37}$$

where $\boldsymbol{X}_{M_N} = LN(MHA(K, Q, V, N) + \boldsymbol{X}_{E_{N-1}})$ and $K = Q = V = \boldsymbol{X}_{E_{N-1}}$ for $N > 1$.

## A.2 DECODER ARCHITECTURE IN THE BASELINE TRANSFORMER

Given a sequence $\boldsymbol{y} = \begin{bmatrix} y_1 & y_2 & \cdots & y_n \end{bmatrix} \in \mathbb{R}^n$ with $n$ elements, the output of a token embedding is a $n \times m$ matrix and it can be written as

$$\boldsymbol{Y}_e = \begin{bmatrix} \boldsymbol{e}^{(1)} & \boldsymbol{e}^{(2)} & \cdots & \boldsymbol{e}^{(n)} \end{bmatrix}^T \boldsymbol{W}_4,$$

where $\boldsymbol{W}_4 \in \mathbb{R}^{|Vocab| \times m}$ is an embedding matrix, $|Vocab|$ is the number of tokens in the vocabulary set, $m$ is the dimension of an embedding vector, and $\boldsymbol{e}^{(i)} \in \mathbb{R}^{m \times 1}$ is a unit column vector with a 1 at position $i$ and 0 elsewhere. The input for the first decoder block is

$$\boldsymbol{Y}_I = \frac{1}{\sqrt{m}}\boldsymbol{Y}_e + \boldsymbol{P}. \tag{38}$$

For each decoder block, it contains a masked multi-head attention layer, a multi-head cross attention, and a feed forward network layer. For the masked multi-head attention layer in the $u$-th decoder block, given three matrices $\boldsymbol{Q}_2, \boldsymbol{K}_2, \boldsymbol{V}_2 \in \mathbb{R}^{n \times m}$, the output is computed by

$$MMHA(\boldsymbol{Q}_2, \boldsymbol{K}_2, \boldsymbol{V}_2, u) = \begin{bmatrix} \boldsymbol{H}_{masked_1} & \boldsymbol{H}_{masked_2} & \cdots & \boldsymbol{H}_{masked_p} \end{bmatrix} \boldsymbol{W}_{masked}^{O_u}, \tag{39}$$

where

$$\boldsymbol{H}_{masked_k} = softmax(\boldsymbol{M} + \frac{\boldsymbol{Q}_2 \boldsymbol{W}_k^{Q_2}(\boldsymbol{K}_2 \boldsymbol{W}_k^{K_2})^T}{\sqrt{m}})\boldsymbol{V}_2 \boldsymbol{W}_k^{V_2} \tag{40}$$

$u = 1, 2, \cdots, N$, $k = 1, 2, \cdots, p$, $N$ is the total number of decoders, $p$ is the number of heads, $\boldsymbol{W}_{masked}^{O_u} \in \mathbb{R}^{qp \times m}$, $\boldsymbol{W}_k^{Q_2} \in \mathbb{R}^{m \times r}$, $\boldsymbol{W}_k^{K_2} \in \mathbb{R}^{m \times r}$, $\boldsymbol{W}_k^{V_2} \in \mathbb{R}^{m \times q}$, $\boldsymbol{M} \in \mathbb{R}^{n \times n}$ is a matrix with

$$\boldsymbol{M}_{i,j} = \begin{cases} \infty & \text{if } j > i \\ 0 & \text{otherwise} \end{cases} \text{ for } 1 \leq i \leq n \text{ and } 1 \leq j \leq n, r \text{ is the dimension of the query and key}$$

projection, $q$ is the dimension of the value projection, $k = 1, 2, \cdots, p$, and $softmax : \mathbb{R}^{n \times s} \to [0, 1]^{n \times s}$ for $s \in \mathbb{N}^+$ is a Softmax function with $softmax(\boldsymbol{A})_{i,j} = \frac{\boldsymbol{A}_{i,j}}{\sum_{j=1}^s e^{\boldsymbol{A}_{i,j}}}$ for $\boldsymbol{A} \in \mathbb{R}^{n \times s}$.

For the multi-head cross attention layer in the $u$-th decoder block, given three matrices $\boldsymbol{Q}_3, \boldsymbol{K}_3, \boldsymbol{V}_3 \in \mathbb{R}^{n \times m}$, the output is computed as

$$MHCA(\boldsymbol{Q}_3, \boldsymbol{K}_3, \boldsymbol{V}_3, u) = [\boldsymbol{H}_1 \quad \boldsymbol{H}_2 \quad \cdots \quad \boldsymbol{H}_p] \boldsymbol{W}_{causal}^{O_u}, \tag{41}$$

where

$$\boldsymbol{H}_k = softmax(\frac{\boldsymbol{Q}_3 \boldsymbol{W}_k^{Q_3} (\boldsymbol{K}_3 \boldsymbol{W}_k^{K_3})^T}{\sqrt{m}}) \boldsymbol{V}_3 \boldsymbol{W}_k^{V_3}, \tag{42}$$

$u = 1, 2, \cdots, N$, $k = 1, 2, \cdots, p$, $N$ is the total number of encoders, $p$ is the number of heads, $\boldsymbol{W}_{causal}^{O_u} \in \mathbb{R}^{qp \times m}$, $\boldsymbol{W}_k^{Q_3} \in \mathbb{R}^{m \times r}$, $\boldsymbol{W}_k^{K_3} \in \mathbb{R}^{m \times r}$, $\boldsymbol{W}_k^{V_3} \in \mathbb{R}^{m \times q}$, $r$ is the dimension of the query and key projection, and $q$ is the dimension of the value projection.

The output of the first decoder block can be written as

$$\boldsymbol{Y}_{D_1} = LN(FFN(\boldsymbol{Y}_{C_1}) + \boldsymbol{Y}_{C_1}), \tag{43}$$

where $\boldsymbol{Y}_{C_1} = LN(MHCA(\boldsymbol{Y}_{M_1}, \boldsymbol{X}_{E_N}, \boldsymbol{X}_{E_N}, 1) + \boldsymbol{Y}_{M_1})$, $\boldsymbol{Y}_{M_1} = LN(MMHA(\boldsymbol{Y}_I, \boldsymbol{Y}_I, \boldsymbol{Y}_I, 1) + \boldsymbol{Y}_I)$, and $\boldsymbol{X}_{E_N}$ is the output of the last encoder block defined in (37).

The output of the $N$ stacks of decoder blocks can be written as

$$\boldsymbol{Y}_{D_N} = LN(FFN(\boldsymbol{Y}_{C_N}) + \boldsymbol{Y}_{C_N}), \tag{44}$$

where $\boldsymbol{Y}_{C_N} = LN(MHCA(\boldsymbol{Y}_{M_N}, \boldsymbol{X}_{E_N}, \boldsymbol{X}_{E_N}, N) + \boldsymbol{Y}_{M_N})$, and $\boldsymbol{Y}_{M_N} = LN(MMHA(\boldsymbol{Y}_{D_{N-1}}, \boldsymbol{Y}_{D_{N-1}}, \boldsymbol{Y}_{D_{N-1}}, N) + \boldsymbol{Y}_{D_{N-1}})$.

