# OpenReview forum: "Efficient transformer with reinforced position embedding for language models"
_ICLR.cc/2025/Conference — ICLR 2025 Conference Withdrawn Submission_

### Official Review · Reviewer_1Sm4 · 2024-10-30

**Soundness:** 2
**Presentation:** 2
**Contribution:** 2
**Rating:** 3
**Confidence:** 5

**Summary:**

The positional encoding is attached to the token embeddings after batch normalization along the feature dimension. To accommodate the residual connections, most weight matrices in the model are doubled in size (except for W_v). The model was trained on a translation dataset, and its performance was compared with the traditional transformer structure through loss comparison.

**Strengths:**

This work presents two interesting counter-intuitive points:

1)	There has been extensive academic research demonstrating that deep-narrow networks perform better than shallow-wide networks, particularly in the translation field. However, this paper essentially doubles the feature width while reducing the number of layers, yet achieves better results.
2)	The rationale for directly adding positional encodings and token embeddings in vector space has been discussed for many years, and remains the mainstream approach even today. However, this paper suggests that separating these spaces is necessary.

**Weaknesses:**

1)	Most of the cited works in this paper are from 2017/2018, and there is a noticeable lack of citations. This raises concerns about whether the authors have a thorough understanding and have conducted comprehensive research in this field. To be honest, this work's issue goes beyond simply needing a few more citations - the paper only has 5 references total, which is far too sparse. While I need to provide some suggestions, I would recommend looking into works in the field of positional encoding, or research analyzing embeddings. I'm afraid I can't offer much more advice to the author beyond that.
2)	The experiments are not solid, with few experimental trials and superficial analysis. It only compares the advantages over the standard transformer model in a very toy setting. I suggest that this work should be combined with more efforts to improve relative/absolute positional encoding, and increase the model's parameter count to enhance soundness. Additionally, conclusions should not be drawn solely from one translation dataset - the effectiveness of this architectural modification should be evaluated on a broader range of tasks including QA, summarization, information extraction, etc.
3)	The structure of the paper is poorly organized. Given that transformer architecture is now very fundamental, making minor modifications only requires brief explanations or could be placed in the appendix - dedicating two and a half pages in the main text is excessive. This leads to a lack of ablation studies, such as analyzing the impact of applying batch normalization to token embeddings and expanding feature dimensions on performance. The paper should include more feasibility analysis from the perspective of vector characteristics. Since there has long been an established understanding that various embeddings can be directly added together, this paper's counter-intuitive approach should dedicate more space to explaining its feasibility. I suggest adding a Related Work section, condensing the lengthy equations and oversized figures in the Methods section, and including an ablation analysis chapter in the Experiments section.
Writing errors or typos:
1)	'Previous work by Ke et al. Ke et al. (2021)' in L33
2)	The order of Figures 1 (b) and 1 (c) has been wrong reversed in L41-42
3)	The model parameter configurations from L300 to L307 were written messily, maybe put it in a table.
4)	'training losses of the baseline have the mean of roughly 6.60, 4.55, 3.82,3.29, 2.89, 2.57, 2.30, 2.11, 1.96, and 1.84 and the mean of the validation losses of the baseline are roughly 5.04, 4.06, 3.46, 3.01, 2.77, 2.49, 2.35, 2.25, and 2.18 for 10 different trials. '  from L315-322,L339-L356. It is not recommended to present such a large amount of numerical results in the main text/figure captions
5)	In Table 1, the three-line table format is not used.

**Questions:**

1)	Why is the model scale so small? The baseline is 10M parameters and the proposed method uses 3M parameters, yet it mentions using an A100 40G GPU. For the experiments to be convincing, the model scale should be at least in the billions of parameters - this would better match both the GPU resources mentioned and current research requirements.
2)	While the paper references Ke et al.'s work, it appears to lack comprehensive comparisons with other works on improving Transformer positional encoding. Should experimental comparisons with other related methods be added?
3)	Would this work be useful for more modern positional encodings, such as ROPE  and ALIBI?

---

### Official Review · Reviewer_tVwQ · 2024-11-02

**Soundness:** 2
**Presentation:** 2
**Contribution:** 2
**Rating:** 3
**Confidence:** 4

**Summary:**

The paper proposes modifications to the standard Transformer architecture by introducing reinforced positional embeddings and alterations in the attention mechanism.
The author propose to concatenate token embeddings with positional embeddings before the first
encoder and decoder blocks, and uses the normalized token embedding matrix as the value in the
attention layer.
Experimental results demonstrates improvements in training loss, validation loss, and computational time
across thirteen translation datasets

**Strengths:**

Give the author credit to carry out extensive results across 14 diverse translation datasets, and shows promising results over the baseline.

**Weaknesses:**

1.The presentation of section 2.3 and 2.4 is a little confusing. And the notation is not quite consistent, making it hard follow. So I feel the proposed modifications to the Transformer architecture are not entirely clear from the provided content. The paper needs to provide a more thorough explanation of how the reinforced positional embedding works and how it interacts with the Transformer's attention mechanism. Additionally, the mathematical formulation and the pseudo-code or diagrams could be more explicit to help reviewers and readers replicate the study.

2.Lack of related work comparison, I think more related work comparison should be added to show the merits of this work, not just compare to baseline transformer.

3. The English writing could be improved. There are some very long sentences which is hard to read, e.g. the second sentence in abstract, maybe the author could chop it into smaller sentences.

**Questions:**

See weakness part

---

### Official Review · Reviewer_sqJU · 2024-11-03

**Soundness:** 2
**Presentation:** 3
**Contribution:** 3
**Rating:** 3
**Confidence:** 4

**Summary:**

The modified Transformer architecture about position embedding improves training loss, validation loss, and computational time across 13 translation datasets compared to a baseline model. It normalizes token embeddings, concatenates them with positional embeddings, and uses the normalized embeddings in the attention layer. The proposed model achieves lower training and validation losses, with an average training time of 1352.27 seconds per epoch, compared to the baseline's 4297.79 seconds per epoch. These improvements suggest the model's robustness across various translation tasks.

**Strengths:**

1. This paper introduces an efficient transformer architecture employing reinforced positional embedding to achieve superior performance with half the number of encoder and decoder layers. Compared to the baseline model, it attains lower training and validation losses while utilizing only a quarter of the training time.

2. The proposed method is mainly aimed at improving the position embedding layer. There are no similar approaches from previous researchers, so the proposed method is novel.

**Weaknesses:**

The experimental results are not solid enough:

1. The baseline model is not very strong, as the proposed method can also be applied to decoder-only architectures. The authors should also use a more recent large language model, such as LLaMA, as the baseline model to validate the proposed method.

2. The evaluation score is not very reasonable, it is recommended to use task-related evaluation scores, rather than just using loss as the final evaluation metric

3. Some related works are missing. Hope to give a comparison with more work about the improved positional embedding, such as RoPE, Relative Position Representations (https://arxiv.org/abs/2104.09864, https://arxiv.org/pdf/1803.02155)

**Questions:**

1. Why did the authors select the Portuguese-English translation task as the experimental task? Hope to see more mainstream translation tasks, such as English-French, English-German, and English-Chinese.

2. Why is only the loss used as the evaluation metric in the experiment? Although the lower the training loss, the better the model performance, it is still desirable to see a comparison of translation-related evaluation scores, such as BLEU.

3. Why did the authors only use the sine-cosine based positional encodings as the baseline? Hope to give a comparison with more work about the improved positional embedding, such as RoPE, Relative Position Representations (https://arxiv.org/abs/2104.09864, https://arxiv.org/pdf/1803.02155)

4. Why did the authors only choose the translation tasks to validate the proposed method? The baseline model is not very strong, as the proposed method can also be applied to decoder-only architectures. I suggest that the authors verify the effectiveness of the proposed method on some large language models such as LLaMA.

5. The authors use a lot of content to explain the principle of reinforced position embedding, but I don't know why this method can improve the convergence speed of the model ? And why can the model converge to a better solution space than the Transformer baseline model? Adjusting the learning rate or warm-up, or using LayerNorm could also achieve the similar effect, right ?

---

### Official Review · Reviewer_RiVe · 2024-11-04

**Soundness:** 2
**Presentation:** 3
**Contribution:** 2
**Rating:** 3
**Confidence:** 3

**Summary:**

This paper studies how to make transformer architecture more efficient by reinforcing the positional embedding. To achieve this goal, the paper makes three proposals: (1) concatenating the token and positional embedding matrices (instead of adding as usual) (2) normalizing the token embedding matrix across tokens (3) use the normalized token embedding matrix as value in the attention layer. Experiments on a few small MT test sets show that the proposed architecture achieves lower training/validation loss with ~30% of the parameter of the baseline model.

**Strengths:**

1. To the best of my knowledge, there is no research paper formally discussing the comparison between concatenating vs. adding positional embedding to the token embedding. Although there is interest and several informal discussions among the community about it (e.g.: https://www.reddit.com/r/MachineLearning/comments/cttefo/comment/exoz3uy/). Hence, this contribution of this paper may fill some relatively important void in the space.
2. The approach is very straightforward, and the description of the method is mostly easy-to-follow.

**Weaknesses:**

1. My biggest complaints are the evaluation of this paper. Further broken down below:
1.1 The title of this paper states that this is "for language models", yet there is no experiment in a T5-like multi-task language model setup.
1.2 Even the machine translation experiments were not well-done. All experiments are done with toy-scale datasets. Evaluation is limited to perplexity, with no actual task-specific evaluation such as BLEU/COMET.
1.3 I'm not sure why we are not using a comparable hyper-parameter setup between the baseline vs. proposed model. It's impressive to see that you are able to achieve comparable performance with a smaller model, but why not use the same number of encoder/decoder blocks/dimensions etc. and show you can perform better with comparable number of parameters?
2. The proposal should also be benchmarked against more up-to-date baselines such as RoPE (https://arxiv.org/abs/2104.09864).
3. The paper lacks a survey of the existing literature.

**Questions:**

More of two detailed comments rather than questions:

- Why is (13) written as an addition inside the softmax? Isn't it easier to formulate it as a binary multiplicative mask outside the softmax?
- The introduction of the architecture could be more concise and focus more on the difference from the baseline. For example, I'm not sure if there's any difference from the baseline that has been introduced in Section 2.4.

---

### Note · Authors · 2024-11-25

I have read and agree with the venue's withdrawal policy on behalf of myself and my co-authors.